# The Role of Dendritic Cells during Physiological and Pathological Dentinogenesis

**DOI:** 10.3390/jcm10153348

**Published:** 2021-07-29

**Authors:** Angela Quispe-Salcedo, Hayato Ohshima

**Affiliations:** 1Department of Tissue Regeneration and Reconstruction, Division of Anatomy and Cell Biology of the Hard Tissue, Niigata University Graduate School of Medical and Dental Sciences, Niigata 951-8514, Japan; aquispesa@dent.niigata-u.ac.jp; 2Department of Stomatology, Faculty of Health Sciences, School of Stomatology, Universidad Cientifica del Sur, Lima 15067, Peru

**Keywords:** cell differentiation, dental pulp, dendritic cells, dentin, extracellular matrix proteins, histocompatibility antigens class II, macrophages, odontoblasts, stem cells, tooth injuries

## Abstract

The dental pulp is a soft connective tissue of ectomesenchymal origin that harbors distinct cell populations, capable of interacting with each other to maintain the vitality of the tooth. After tooth injuries, a sequence of complex biological events takes place in the pulpal tissue to restore its homeostasis. The pulpal response begins with establishing an inflammatory reaction that leads to the formation of a matrix of reactionary or reparative dentin, according to the nature of the exogenous stimuli. Using several in vivo designs, antigen-presenting cells, including macrophages and dendritic cells (DCs), are identified in the pulpal tissue before tertiary dentin deposition under the afflicted area. However, the precise nature of this phenomenon and its relationship to inherent pulp cells are not yet clarified. This literature review aims to discuss the role of pulpal DCs and their relationship to progenitor/stem cells, odontoblasts or odontoblast-like cells, and other immunocompetent cells during physiological and pathological dentinogenesis. The concept of “dentin-pulp immunology” is proposed for understanding the crosstalk among these cell types after tooth injuries, and the possibility of immune-based therapies is introduced to accelerate pulpal healing after exogenous stimuli.

## 1. Introduction

The dental pulp is a specialized soft connective tissue of ectomesenchymal origin harboring distinct cell populations with specific functions involved in teeth production and maintenance [1,2,3]. Odontoblasts—dentin-forming cells—constitute the most specialized and important cells of the tissue [4,5,6], whereas fibroblasts—the most numerous cells of the pulp—are responsible for the synthesis and maintenance of the pulp extracellular matrix (ECM). Other important cell types include undifferentiated mesenchymal stem cells (MSCs)/progenitor cells, blood vessel components, nervous cells, and a wide variety of defense cells [1,2,3,7]. Under physiological conditions, the pulpal tissue executes self-regulatory mechanisms that maintain the equilibrium of these cell populations [8,9]. Previous studies in murine and human teeth demonstrated that the dental pulp has low proliferative activity, except for the apices of developing roots or the apical end of incisors in rodents [2,7,10,11]. Likewise, other studies have proven that apoptosis, or programmed cell death, occurs at low rates in the normal dental pulp of continuously growing incisor in rats and in the pulps of human mature teeth throughout life, regulating the turnover of odontoblasts, cells of the subodontoblastic layer, and fibroblasts [2,11,12,13,14,15]. Therefore, given the stable characteristics of the dental pulp under physiological conditions, it is essential to understand the tissue to the extent of its responsiveness after exogenous stimuli to the tooth.

Exogenous transdentinal stimuli, such as dental carious lesions, abrasion, attrition, dental traumatisms and restorative procedures, induce harmful variations in the odontoblast layer at the afflicted site, eliciting an immunocompetent response in the pulpal tissue [7,16,17,18,19]. Odontoblasts are considered as the first line of defense against pathogen invasion, facilitating the initiation, development, and maintenance of the immune/inflammatory response. The release of proinflammatory cytokines by the afflicted odontoblasts triggers specific intracellular signaling pathways involving nuclear factor-κB and mitogen-activated protein kinase p38 that promote the recruitment of leukocytes and antigen-presenting cells (APCs), including macrophages and immature dendritic cells (DCs) [4,5,6,20,21,22]. After the activation of the pulpal cellular network, a newly formed pathological dentin matrix is steadily deposited beneath the injury site. This calcified matrix, also known as tertiary dentin, can be further classified as either reactionary or reparative dentin based on the differences in the exogenous stimuli intensity to the tooth and the biological events in the pulpal tissue. Thus, the tertiary dentin matrix secreted by surviving odontoblasts in response to a mild stimulus is defined as reactionary dentin; whereas reparative dentin is defined as the matrix deposited beneath the afflicted area by newly differentiated odontoblast-like cells after the death of the original odontoblasts due to severe injuries [7,23,24,25,26,27,28,29,30,31,32,33]. The latter implies a more complex process involving important biological mechanisms, such as apoptosis, cell proliferation, and cell differentiation, which remain to be elucidated at cellular and gene levels [34,35,36,37,38]. APCs are also critical during the initial defense reaction. Previous in vivo studies have shown various patterns for these cells at the onset of the healing process. For instance, macrophages are located in the central portion of the pulp, whereas immature DCs can be detected in the odontoblast layer [7,16,39,40,41,42,43,44,45,46,47,48,49,50,51]. DCs are sentinels that maintain immune homeostasis, orchestrating the immune system’s components for a favorable effect in an organism [52,53]. Besides its main role in the host-response against pathogens, the paraodontoblastic location of these cells may suggest another important role during pulpal healing that needs to be clarified.

In the last decades, several animal models mimicking external injuries to the teeth have been established to analyze the behavior of distinct pulpal cell populations, with special emphasis on the crosstalk of dental pulp stem cells (DPSCs) and other inherent cells. Study designs using in vivo approaches included autogenic/allogenic tooth germ and whole-tooth replantation and transplantation [35,36,49,54,55,56,57,58,59], cavity preparations with or without pulp exposure [16,37,38,44,47,48,60,61], and root resection [62]. For instance, the injection of 5-bromo-2′-deoxyuridine in young animals at the optimal time demonstrated the localization of slow-cycling long-term label-retaining cells (LRCs). It was observed that LRCs in transplanted teeth maintain their proliferative and differentiation capacities despite extensive apoptosis occurring in the pulpal tissue of the transplant and play crucial roles in the pulpal healing process after exogenous stimuli, leading to the conclusion that dense LRCs are believed to be dental stem/progenitor cells in mature pulp tissue [63,64,65]. Therefore, animal models simulating different kinds of tooth injuries are necessary to identify the behavior of DPSCs in relation to other cellular lineages of the pulpal tissue, such as APCs [66,67,68]. Hence, this literature review discusses the possible role of dental pulp DCs and their relationship to odontoblasts, DPSCs, and other resident cells involved during physiological and pathological dentinogenesis.

## 2. Key Functions of DCs

DCs are professional APCs critical for the initiation and orchestration of the immune response [69]. DCs were described for the first time in 1973 by Dr. Ralph Steinman (Nobel Prize of Medicine and Physiology in 2011) as a heterogenous population of leukocytes originated from hematopoietic stem cells different from macrophages [52,70]. DCs are found in most parts of the human body, including the lymph nodes, skin, blood, spleen, lungs [71], and oral tissue [39,42,72], where they play a key role in maintaining immune homeostasis by both activating adaptive immunity and contributing to tolerance [53]. DCs can be classified into different subsets with specialized functions in immune responses to specific pathogens [73]. The two major DC subsets are plasmacytoid DCs (pDCs) and myeloid/conventional DCs (cDCs). These subsets are generally identified based on the lack of expression for T cells, B cells, natural killer (NK) cells, or other granulocyte-specific markers, high-level expression of major histocompatibility complex (MHC) class II, and lack of monocyte markers [69,74]. pDCs play an important role in the innate immune system because of their capacity to produce type I interferons upon viral infection and are also involved in tolerance or immune suppression in their immature state [75]. Differently, cDCs recognize bacterial components and produce proinflammatory cytokines to activate proinflammatory T-cell subsets and subsequently promote the recruitment of cytotoxic T lymphocytes. cDCs can be further divided into cDC1 and cDC2. Human cDC1s and cDC2s can be found in the blood and lymphoid and nonlymphoid tissue [75,76]. Thus, cDCs constitute a distinct lineage of cells with the capacity to seed tissue and maintain immune homeostasis in a steady state while rapidly responding to local insults and initiating and directing innate and adaptive immunity [76]. Immature cDCs accumulate at the inflammation site along with other phagocytic cells (e.g., macrophages) in response to the release of chemokines in this area. Antigens and their associated molecules will be later recognized and captured by cDCs through different mechanisms, including pinocytosis (macropinocytosis and micropinocytosis), phagocytosis, and receptor-medicated endocytosis. Subsequently, cDCs will migrate toward lymphoid tissue during their maturation process, and the antigenic peptides processed by cDCs will be presented on MHC class I and II molecules for interaction with different subsets of T cells [75,77,78]. Because pDCs are not the object of the analysis of this review, but cDCs (particularly those in the dental pulp), cDCs will simply be addressed as DCs.

## 3. Relationship between DCs and Odontoblasts during Physiological Dentinogenesis

The existence of MHC^+^ class II cells with dendritic features in a normal human pulp tissue was first reported in 1987 by Jontell et al. [79]. Cell surface markers, such as human leukocyte antigen (HLA)-DR and HLA-DQ isotypes, and leucine (Leu) 3a were used to identify the presence of putative DCs in human samples. HLA-DR^+^, HLA-DQ^+^, and Leu 3a^+^ cells showed clear dendritic features and were predominately located at the periphery of the dental pulp [79]. Another study in human pulp samples reported similar results. HLA-DR^+^ cells of dendritic appearance formed a reticular network in the pulpal tissue while coexpressing factor XIIIa, a marker for human DCs. DCs were observed in close relation with the endothelial cell membrane, forming a three-dimensional structure around the microvessels [80,81]. Ohshima et al. confirmed previous reports on the characteristics and location of human dental pulp DCs. They summarized the spatial relationship between MHC^+^ class II cells and odontoblasts (Figure 1) [82].

For instance, cytoplasmic extensions of HLA-DR^+^ cells in the predentin were seen in the dentinal tubules, while being in close relationship to several odontoblast processes (Figure 1 and Figure 2b). Pulpal DCs were consistently identified at the periphery of the tissue, inside and beneath the odontoblast layer (Figure 2a). The ultrastructure of HLA-DR^+^ cells evidenced several vesicles, a moderately developed Golgi apparatus, and mitochondria [82]. These results were consistent with other reports on HLA-DR^+^ cells in the dental pulp of human unerupted developing teeth and human deciduous teeth during the physiological root resorption process [81,83].

Kannari et al. reported that in the dental pulp of human deciduous teeth under physiological root resorption, MHC^+^ class II cells were seen in the same location as in previous reports (the odontoblast layer and/or predentin), extending their processes into the dentinal tubules [41]. By transmission electron microscopy (TEM) analysis, these cells also showed the presence of clear and electron-dense vesicles of different dimensions, although the number of cell organelles was scarce (Figure 2c) [82]. Lastly, a recent quantitative analysis of immunocompetent cells of the human dental pulp using fluorescence-activated cell sorting, together with identifying specific cell subsets in leukocyte (CD45^+^) cells, provided that DCs represent 4.51 ± 1.12% of the leukocyte population, thus supporting the data provided by conventional histological approaches [84].

Although studies using human teeth samples during the dentinogenesis process were pioneering and very informative, the data retrieved remained limited. Difficulties in sample acquisition and other ethical concerns led to the setting of murine models for further investigation in this subject and under pathological conditions [66,67,68]. The rodent incisor is a continuously growing tooth that constitutes an optimal model for observing the different stages of dentinogenesis [85]. Different markers are available for identifying and tracking putative DCs regardless of the origin of the sample. Approximately 30% of MHC^+^ class II cells in rats were positive for the rat DC marker OX6 [86]. In contrast, another study on the mouse pulp verified the existence of two subpopulations based on their positive immunoreaction for CD11c^+^ and F4/80^+^ markers [87].

The presence of DCs in the normal pulpal tissue of rat incisors was verified for the first time in 1988 also by Jontell et al. using immunohistochemistry for Ia antigen and TEM. Rat pulpal DCs were observed at the pulp periphery, where they exhibited narrow and tortuous cell processes, well-developed mitochondria, and an intense expression of class II antigen [88]. Similarly, Okiji et al. investigated the heterogeneity of macrophage-like cells and DC-like cells in the normal rat molar pulp by using double immunoperoxidase staining with OX6, ED1 (monocytes, macrophages and DCs), and ED2 (tissue macrophages or histocytes) antibodies. They identified a subpopulation of OX6^+^/ED2^-^ cells at the periphery of the coronal pulp of dendritic appearance [89]. Additionally, another study in rat incisors demonstrated that DCs increased in number according to the progress of dentinogenesis. At early stages, scattered OX6^+^ cells in the dental pulp were seen beneath the odontoblast layer. In contrast, several OX6^+^ cells were found throughout the dental pulp at the site of active dentin formation. These cells were characterized by their dendritic features, showing a close relationship to odontoblasts and the intervening network of fenestrated capillaries close to the predentin (Figure 2d) [39]. As for its intracellular architecture, OX6^+^ cells contained multivesicular bodies, other smaller vesicles with amorphous contents, fine tubulovesicular structures, and a well-developed Golgi apparatus (Figure 2e) [39]. Although DCs and macrophages express MHC class II antigen, DCs differ from macrophages not only for their distinct morphological features but also for their low phagocytic capacity and low acid phosphatase (ACPase) activity. This notion was supported by another in vivo study in rat incisor pulp, where a subpopulation of OX6^+^ cells associated with fenestrated capillaries in the odontoblast layer, at the periphery or central portions of the pulp tissue of rat incisors, demonstrated low levels of ACPase activity, being considered a type of “true” DCs [50]. Consistent with previous findings, pulpal DCs cells were also confirmed in postnatal developing mice and the molars of aged rats. In mouse molar samples, the use of the F4/80 antibody allowed the detection of immature DCs inside and nearby the odontoblast layer during the morphogenetic stage [90]. Similarly, in aged rat molars, several OX6^+^ cells exhibiting dendritic features were seen throughout the dental pulp but mainly in the peripheral zone near the pulp-dentin border. An intense immunoreaction was observed in cellular bodies and processes of OX6^+^ cells in the subodontoblastic and odontoblast layers and those extending into the predentin, respectively (Figure 2f) [47].

## 4. Relationship between DCs and Odontoblasts or Newly Differentiated Odontoblast-like Cells after Noninfected Exogenous Injuries

Exogenous injuries to the tooth elicit the response of the local immune system of the pulpal tissue. The pulpal response directly involves various cell types that are distinctly regulated depending on the intensity and duration of these injuries. Thus, a sequence of biological mechanisms, such as apoptosis, cell proliferation, cell migration, and cell differentiation, is triggered in the pulpal tissue immediately after the injury, leading to proper healing of the dental pulp. The establishment of murine injury models without infection is advantageous to monitor the presence of DCs at the lesion site and their behavior after the injury. By excluding the influence of external harmful agents, such as bacteria and related substances, it is easier to analyze the influence of the local microenvironment for DC recruitment and their relationship to other pulpal cell populations.

The first study that explored the occurrence of pulpal MHC^+^ class II cells under pathological conditions was published in 1991 by Bergenholtz et al. The dental pulp of rat incisors was exposed and irritated with bacteria-derived lipopolysaccharides and then covered with a temporal restoration. A significant increase in MHC^+^ class II cells was observed in inflamed pulps compared to control. Cells presenting dendritic features were more abundant than those observed in the normal pulps of incisors [91]. Another similar study explored the response of immunocompetent cells in relation to neural elements using innervated or denervated pulps of rat molars. Four days after the operation, Ia^+^ cells (a marker for immunocompetent cells) were densely distributed in the proximity of the injury in the innervated group, suggesting the importance of sensory nerves on the recruitment of immunocompetent cells [92].

Murine incisors are continuously-growing teeth and not suitable for comparison to human teeth in terms of healing and/or regeneration. Therefore, injury models were modified using murine molars for cavity preparations with or without pulp exposure instead of incisors. Moreover, the murine tooth replantation model was proposed to analyze the pulpal responses after severe injuries, mimicking a situation where the neural and vascular supply of the pulpal tissue is abruptly interrupted. Thus, the use of noninfection injury models in the past years has helped to clarify the pulpal cell dynamics, allowing to understand the relation among immunocompetent cells, odontoblasts, and odontoblast-like cells.

The preparation of occlusal cavities in rat molars has shown to induce the immediate disruption of odontoblast morphology and cell death, causing a sharp increase in apoptotic activity. The induction of two waves of apoptosis has been reported in odontoblasts after the cavity preparation in murine molars. In rats, the primary induction of the apoptotic activity in odontoblasts appeared just 1 h after the operation, whereas the secondary induction of apoptotic cells appeared in the subodontoblastic layer 1 day after the operation [93]. Differently, cavity preparations in the mouse molar elicited an extensive apoptotic activity from 12 h to day 1 after the operation, which was resolved after day 3. The intense apoptotic activity shown in mice was identified not only in odontoblasts but also in the subodontoblastic layer, expanding toward the center of the pulpal tissue [38]. The differences in the location, wideness of inflammatory reaction, and timing of the apoptotic activity in the pulpal tissue of rats and mice may be explained by the size of their pulp chambers and overall tooth dimensions.

After the initial acute injury, exudative lesions appeared beneath the injury site and separated the few surviving odontoblasts from the predentin. MHC^+^ class II cells, identified by the OX6 antibody, moved inward from the pulp-dentin border, and were relocated beneath the exudative lesions (asterisk) (Figure 3a) [40,44,94]. Six hours after the cavity preparation, the surviving odontoblasts began to degenerate and lost HSP-25 expression (a protein used as a rat odontoblast marker), whereas some of the OX6^+^ cells remained among them. Scattered OX6^+^ cells extended their dendritic process into the dentinal tubules. However, the number of these cells increased 12 h after the operation, especially under the afflicted area (arrow in Figure 3b). Exudative lesions disappeared from 12 h to day 1 after the operation (Figure 3b). Odontoblasts did not show HSP-25 immunoreactivity. Moreover, when samples were analyzed by TEM, the cytoplasmic processes of OX6^+^ cells located deep into the dentinal tubules were characterized by tubulovesicular structures, multivesicular bodies, and vacuoles, as seen in DCs observed in other human tissue [40,44,94]. It is believed that pulpal macrophages and neutrophiles quickly remove the cell and dentinal debris induced by cavity preparation, thus facilitating the migration of DCs toward the pulp-dentin border [93]. Newly differentiated odontoblast-like cells began to appear 48 h after the initial injury. Although speculative, there is a possibility that HSP-25 can be discharged from dying odontoblasts to the extracellular milieu, where it could act as a chemotactic agent, favoring DC recruitment in the afflicted area [40,44,94]. Up to day 3 after the operation, the afflicted area of the pulp completely recovered, and newly differentiated odontoblast-like cells expressing HSP-25 aligned under predentin. OX6^+^ cells were located exclusively beneath the odontoblast layer (Figure 3c) and displayed the same ultrastructural features as those observed in control samples [40,44,94]. Previous findings were also confirmed in aged rat molars after the same injury model. Aged pulps still maintained a satisfactory self-defense capacity, with the same rate of recovery of younger pulps [47,95]. The summary of the morphological changes and the spatiotemporal relationship between odontoblasts and OX6^+^ cells in a rat model for cavity preparation are summarized in Figure 4a.

To analyze the behavior of DCs after severe injuries, another set of studies was carried out using the murine tooth replantation model. Rungvechvuttivittaya et al. investigated the kinetics of pulpal macrophages and MHC^+^ class II cells after the tooth replantation. From days 3 to 7 after the operation, OX6^+^ cells with a dendritic profile concentrated in the pulp-dentin border areas where odontoblasts had died. Two weeks after the operation, a calcified tissue was observed under the predentin in all replanted teeth, indicating that the pulpal repair process had been completed [23,25]. Interestingly, ED1^+^ cells (a marker for macrophages) were seen in the same high proportion as OX6^+^ cells in cases where bone-like tissue was formed in the pulp of the replanted teeth [42]. Ohshima et al. confirmed the previous reports in two conclusive studies after the same methodology in rats. The activity of odontoblasts and odontoblast-like cells was related to the variation of OX6^+^ and ED1^+^ cell distribution in the dental pulp during its repair process. On day 1 after replantation, numerous polymorphonuclear leukocytes (PMLs) migrated through the damaged odontoblast layer, which lost HSP-25 immunoreactivity. OX6^+^ and ED1^+^ cells increased in number at the pulp periphery at this stage. On day 3, odontoblasts did not show HSP-25 immunoreactivity, nor major changes were noted in the number of OX6^+^ and ED1^+^ cells located at the pulp-dentin border (arrows) (Figure 3d). The ultrastructure of these cells presented the same tubulovesicular structures and multivesicular bodies in their cytoplasm as other pulpal DCs, whereas an intense immunoreactivity for OX6 was recognized in their cell membranes (arrows in inset) (Figure 3e). From days 3 to 5, OX6^+^ cells increased in number at the pulp periphery of replanted teeth. Newly differentiated odontoblast-like cells were identified on day 5 after replantation, showing intense HSP-25 immunoreactivity. Some OX6^+^ cells with dendritic profiles were observed beneath or in between them (Figure 3f). OX6^+^ cell accumulation was observed specifically in few areas where the pulpal repair process was still delayed. One week after replantation, reparative dentin was deposited by newly-differentiated odontoblast-like cells in most samples, and the number of OX6^+^ and ED1^+^ cells returned to normal. Scattered OX6^+^ cells remained at the pulp periphery and shifted their location to the subodontoblastic layer. From week 2 to up to 90 days, at least three distinct regeneration patterns were observed in the dental pulp of replanted teeth: reparative dentin, bone-like tissue, and a mixed form of reparative dentin and bone-like tissue. Interestingly, ED1^+^ cell aggregation and the appearance of abundant nerve fibers preceded the bone-like tissue healing pattern [43,45]. The spatiotemporal relationship between odontoblasts and OX6^+^ cells during the early stages of the pulpal defense response after the tooth replantation in rat molars is summarized in Figure 4b.

Although there are some differences between the cavity preparation and tooth replantation models, the results of both models provided valuable information on the possible role of DCs during the pulpal regeneration/repair process. The replantation procedure provoked an abrupt interruption of the neural and vascular supply, inducing a low-oxygen environment that delayed the biological mechanisms driving the pulpal defense response. Odontoblasts slowly degenerated due to the progressive lack of oxygen, which delayed the initiation of the apoptotic activity. Therefore, if the initiation of the apoptotic activity is delayed in the pulpal tissue of replanted teeth, pulpal DC aggregation and the subsequent steps, including odontoblast-like cell differentiation, will also be delayed. This constitutes one of the major reasons to explain the timing of odontoblast-like cell differentiation in both models. Instead, the injury provoked by the preparation of cavities causes a direct impact on odontoblasts, leading to a faster progression of the pulpal healing process. Besides, other intervening factors, such as the release of growth factors from the preexisting dentin and the increased proinflammatory cytokine levels, could also influence the changes in the timing of appearance of OX6^+^ cells and the newly-differentiated odontoblast-like cells between the cavity preparation and tooth replantation models.

## 5. Responses of DCs to Bacterial Infection in the Dental Pulp

In established deep carious lesions, oral pathogens and related molecules effectively reach the pulp-dentin interface through the exposed dentinal tubules, causing a severe inflammatory reaction. The afflicted odontoblasts steadily release chemokines aiming to recruit local immune cells, such as macrophages and DCs, at the exposure site to initiate both innate and adaptive defense responses in the pulpal tissue [96,97]. Kamal et al. established for the first time an experimentally-induced caries model in rat molars. Experimental animals were orally inoculated with *Streptococcus mutans* and subjected to a controlled cariogenic diet. The initial pulpal response was characterized by a marked but transient OX6^+^ cell accumulation, showing dendritic features localized in and around the odontoblast layer, especially under the exposed area. Additionally, when reparative dentin was observed in the teeth, the number of OX6^+^ cells was considerably reduced. However, when the reparative dentin was invaded by caries, OX6^+^ cell accumulation reappeared [98]. In human samples, few studies have addressed the dynamics of immunocompetent cells in vivo. Izumi et al. described HLA-DR cells in the pulp of intact and carious teeth at different stages of infection. In the early stages of carious lesions, a small number of HLA-DR^+^ cells were observed under the affected area, whereas they increased according to the progression of the carious lesions [99]. Similarly, another study reported the behavior of HLA-DR^+^ dendritic-like cells in the dental pulp of teeth with established carious lesions. In early-stage carious lesions, HLA-DR^+^ cells were recruited under the subodontoblastic layer, whereas, in severe carious lesions, positive cells expanded along the odontoblast layer. In the most severe situations, HLA-DR^+^ cells migrated toward the core of the pulp. Moreover, the presence of HLA-DR^+^ cells was lower in carious teeth with reparative dentin than those with active lesions [81]. Interestingly, the occurrence of nerve fibers and DCs in the pulpal tissue under carious lesions showed a synchronized density and distribution according to the severity of the lesions. These observations may suggest an unexplored neuroimmune interaction contributing to the initial response and modulation of the pulp-dentin complex [100,101]. Another study has proven that the cavity depth alters the HLA-DR^+^ DC distribution. A half-reduction of the dentin thickness directly changed the DC distribution, whereas a two-third reduction directly induced the replacement of odontoblasts with HLA-DR^+^ cells until 1 month after treatment. Clusters of immunocompetent cells, such as DCs and T lymphocytes, and nerve fibers remained in the subodontoblastic layer under the affected area 6 months after treatment [83,102].

Other infection models focused on the distribution and fate of MHC^+^ class II cells in rat molars considering a variation in experimental models, such as the use of laser ablation for cavity preparation, and their treatment with temporal fillings, including the use of a combination of antibacterial drugs for the in situ disinfection of oral-exposed cavities [46,48]. For instance, cavity preparation with chromium, thulium, erbium: yttrium-aluminum-garnet (CrTmEr:YAG) laser causes the immediate appearance of exudative lesions and the separation of odontoblasts from the pulp-dentin border. OX6^+^ cells showing a dendritic appearance at the affected site shifted inward together with the separated and damaged odontoblasts. From 6 to 12 h after the cavity preparation, exudative lesions seemed to disappear, and numerous OX6^+^ cells appeared along the pulp-dentin border, extending their processes deep into the exposed dentinal tubules of nontreated cavities. One day after the operation, inflammatory cells rapidly increased in number and were recruited between the predentin and the damaged or survival odontoblasts. OX6^+^ cells shifted their location from the pulp-dentin border to around the inflammatory lesions. These cells lost their cytoplasmic processes and appeared rounded. Laser ablation induced the easy infiltration of PMLs to form an abscess lesion in the dental pulp in most samples of the nontreated group at days 3–5 after the operation. 

Scattered OX6^+^ cells were observed at a distance from the inflammatory abscess in the dental pulp (Figure 5b). At higher magnification, the penetration of masses of oral bacteria was identified in dentinal tubules beneath the untreated cavity (Figure 5c,d). In contrast, samples sealed with temporal cement after the cavity preparation showed no abscess formation in the dental pulp, coinciding with the results of a similar study using the same laser type [103]. Few OX6^+^ cells were identified along the pulp-dentin border under the affected area. At higher magnification, these cells extended their cellular processes into the dentinal tubules, although the penetration was slight compared to the group without treatment (Figure 5a). The findings of this study suggested that bacterial infection and subsequent abscess formation impaired DC recruitment, which may have contributed to the delay in the pulpal repair/regeneration process [46]. To further analyze the influence of bacterial contamination on DC recruitment in the afflicted pulpal tissue, Sato et al. introduced the use of an α-tricalcium phosphate (αTCP) cement containing antimicrobials, such as ciprofloxacin, metronidazole, and cefaclor (3Mix). The study aimed to clarify the responses of neural and immune cells to antimicrobials during the healing process of infected pulps in a rat cavity preparation model. After exposure to the oral environment for 12–24 h, the exposed cavities were covered with αTCP cement (control group) or αTCP cement containing 3Mix (experimental group), followed by restoration with glass ionomer cement. In the control group, large abscess lesions, including numerous neutrophils, were observed from day 3 to week 2 in the mesial half of the coronal pulp (Figure 5e). Neutrophil accumulation was observed at the exposed area covered with αTCP cementum, whereas OX6^+^ cells or PGP 9.5^+^ nerves were not consistently seen around the abscess lesions (Figure 5f). In contrast, numerous OX6^+^ cells accumulated along the pulp-dentin border of cavities treated with αTCP cement containing a 3Mix drug combination (Figure 5g). In situ disinfection induced by the mixture of antibiotics accelerated tertiary dentin deposition (Figure 5h) by newly differentiated odontoblast-like cells located beneath the newly formed dentin matrix (asterisk in Figure 5h) two weeks after the injury. PGP 9.5^+^ nerve fibers were densely distributed in the coronal pulp, whereas OX6^+^ cells retreated beneath the odontoblast-like cells (arrows in Figure 5h). Statistical difference in the number of OX6^+^ cells was found between the experimental and control groups [48].

The findings obtained using infection models demonstrate the importance of DCs as initiators of the pulpal immune response and suggest another putative role during the pulpal healing process [46,48,98,103]. The pulpal immune responses may vary according to the local conditions and the stimuli intensity, as the antigen-presenting capacity of pulpal DCs may persist for many months, even after caries treatment [83,100,102]. During the early stages after oral exposure, an active DC recruitment is described in the dental pulp under the exposure site [81,99]. Once an abscess lesion is formed, numerous neutrophils with phagocytotic activity occupy the afflicted odontoblast layer, whereas the remaining DCs migrate toward the center of the pulp and/or the surrounding areas. Changes in DC distribution may be related to the normal transit from the infection site to the regional lymphatic nodes for their maturation and the activation of the innate tissue-specific response of the dental pulp. This phenomenon does not occur if the infection is prevented by either covering the exposed cavity or using special filling materials, such as αTCP cement containing 3Mix drugs, for the in situ disinfection of the pulpal microenvironment. Although the signaling pathway connecting odontoblast-like cells and pulpal DCs is still unknown, it is evident that the interplay of odontoblasts and immunocompetent cells influences the healing of the pulpal tissue even under infection conditions, as all presented studies have shown that the number of OX6^+^ cells consistently decreased after tertiary dentin deposition [46,48,81,83,96,97,98,99,100,101,102,103]. Further research should confirm whether DCs play dual roles in both the initial immune response and the differentiation process of odontoblast-like cells under pathological conditions.

## 6. Relationship between DCs and Osteopontin (OPN) during Pathological Dentinogenesis

OPN is a complex cytokine and adhesion protein that contains an integrin-binding RGD (arginine-glycine-aspartic acid) sequence commonly found in ECM molecules. OPN is encoded by a single copy gene located in the SIBLING (small integrin-binding ligand, N-linked glycoproteins) cluster on human chromosome 4 and mouse chromosome 5 [104,105]. Although OPN was originally identified as a sialoprotein produced by osteoblasts, it is now proven that a wide variety of cell types such as endothelial cells, smooth muscle cells, epithelial cells, odontoblasts, and immunocompetent cells including activated macrophages, NK cells, and DCs, also produce OPN in response to various stimuli [106,107,108,109]. Therefore, the role of OPN is critical for fundamental physiological functions, such as tissue remodeling, cellular immune responses, and calcium homeostasis in milk and urine. However, in inflamed and injured tissue, OPN is strongly upregulated and involved in the pathogenesis of various inflammatory disorders, such as autoimmune disorders, several cancers, and cardiovascular diseases [104,110].

In dental tissue, OPN constitutes one of the fundamental non-collagenous dentin matrix proteins. It is identified in odontoblasts during dentinogenesis and in the predentin of erupted teeth [108]. Besides its role during physiological dentinogenesis, OPN participates in initiating the pulpal reparative process after tooth injuries. Previous studies have identified OPN expression on the boundaries between the predentin and the newly-formed dentin matrix in the transplanted tooth [56]. Also, OPN expression was found between the necrotic matrix and reparative dentin after the direct capping of mineral trioxide aggregates on the mechanically exposed dental pulp [60].

OPN has anti-inflammatory actions, as it regulates innate and adaptive immune responses [110,111,112]. Human macrophages and DCs synthesize OPN, contributing to the differentiation, maturation, activation, and survival of DCs, through autocrine and/or paracrine pathways. In macrophages, OPN is important for the migration of these cells and plays an important role in phagocytosis and bacterial cell killing [104,109,113]. Likewise, the presence of other molecules in the local microenvironment, such as the granulocyte macrophage colony stimulating factor (GM-CSF) and M-CSF, is also known to induce DC and macrophage differentiation from the common myeloid precursor cells, respectively [114].

In the heavily afflicted tooth, the local microenvironment plays a major role in determining the healing patterns in the dental pulp. In a detailed in vivo study, Saito et al. analyzed the influence of intervening molecules from the pulpal microenvironment, such as GM-CSF and OPN, in the reparative dentin formation process by tracking the differentiation process of odontoblast-like cells. They used a modified allogenic tooth transplantation model, where only the coronal portion of the tooth, without periodontal tissue, was transplanted into the sublingual region of mice. The sequence of cellular events was analyzed by immunohistochemistry for nestin (an odontoblast marker), OPN, MHC class II, and GM-CSF and in situ hybridization was performed for OPN. On day 1, the pulp cavity was mainly occupied by inflammatory lesions, and some cells that extended their cellular processes into the dentinal tubules showed GM-CSF^+^ reactions. OPN^+^ reactions were recognized in the fibrin networks of the pulp cavity and the pulp-dentin border, whereas *Opn* mRNA^+^ cells were scattered spotted in the pulpal tissue. On day 3 after transplantation, double immunofluorescent staining showed that MHC^+^ class II cells with dendritic features, located along the pulp-dentin border, were also positive for OPN, indicating that OPN^+^ cells were immunocompetent cells, such as macrophages and DCs (Figure 6a). Intense OPN^+^ reactions were seen only at the mineralization front of the preexisting dentin on day 5, *Opn* mRNA^+^ cells were recognized at the pulp-dentin border on days 5 and 7 (Figure 6b,c), and nestin-positive newly differentiated odontoblast-like cells were arranged along the pulp-dentin border where OPN^+^ cells had previously existed (Figure 6d). On day 7, tubular dentin formation began to be deposited next to the preexisting dentin. Interestingly, GM-CSF expression began to disappear from the pulp-dentin border and dentinal tubules on days 5-7, except for the sites where inflammatory reactions remained. Nevertheless, GM-CSF^+^ cells reappeared beneath the differentiated odontoblast-like cells from day 5 to week 2. Finally, 2 weeks after the operation, sustained OPN immunoreactions were observed at the boundary between the predentin and the newly-formed dentin matrix. OPN^+^ osteoblasts surrounded the bone matrix (Figure 6e). In summary, the results of this study showed that GM-CSF and OPN secretion by pulpal macrophages and DCs is important for pulpal DC maturation and the subsequent acceleration of the odontoblast-like cell differentiation process.

In this model, transplantation involved the complete excision of the tooth out of its alveolar socket inducing a low oxygen environment in the remaining pulpal tissue. It was previously reported that hypoxia significantly increases OPN production in immature and mature DCs with critical implications for tumor pathogenesis and inflammatory and autoimmune diseases [104,115]. In addition, other studies highlighted that OPN might be a fundamental factor for cell survival regardless of cell lineage, preventing apoptosis. Thus, the role of locally synthesized OPN in DC survival appears to be beneficial for maintaining homeostasis at the inflammation site [109]. The release of growth factors from various sources, such as the preexisting dentin matrix, inflammatory cells, or autolytic odontoblast and pulpal cells, may favor DC recruitment at the injury site; subsequently, the odontoblast differentiation process will proceed steadily [7]. Although further experimental data are needed, it can be hypothesized that OPN secretion during the odontoblast differentiation process by pulpal macrophages and DCs may induce positive feedback not only for immature DC maturation but also by preventing them from extensive apoptosis [109].

## 7. Relationship between DCs and Stem/Progenitor Cells in the Dental Pulp after Injuries

Stem cells of the dental pulp are called DPSCs or, in immature teeth, stem cells from human exfoliated deciduous teeth. Other similar cells include dental pulp from the apical papilla and dental follicle progenitor cells. These cell populations have the capacity to differentiate into odontogenic cells and other cell lineages [7]. Bone marrow resident MSCs are located in perivascular areas of the bone marrow microenvironment, where they can be in close contact with immune cells, including B cells, T cells, and DCs [104,116]. Likewise, MSCs in the dental pulp harbor in perivascular niches, where they display a phenotype consistent with pericytes. It was proven that perivascular DPSCs are directly involved in the regulation of the tissue regeneration after mild or severe injuries, as in the formation of reactionary or reparative dentin [117,118,119]. Therefore, the ability of both young and old teeth to respond to injury by the induction of pathological dentinogenesis suggests that a small population of competent progenitor cells or pulp stem cells may exist within the dental pulp throughout life [7].

Previous in vitro experiments demonstrated that MSCs in a steady state exhibit inhibitory effects on DC differentiation, which resemble immature/semimature/tolerogenic DCs, even upon exposure to pro-maturation stimuli. In contrast, targeted downregulation of suppressive mechanisms by which MSCs act on DC differentiation and functions could break the immune tolerance in immunosuppressive environments such as tumors [120]. In addition, it has been proven that resting MSCs can further increase OPN production when cocultured with DCs. In contrast, in the presence of proinflammatory cytokines, MSCs exert an opposite effect inhibiting OPN production. OPN production by DCs and DC-conditioned medium enhances the osteogenic differentiation of MSCs, leading to the upregulation of the osteogenic markers alkaline phosphatase (ALP) and RUNX2 and the expression of the bone-anabolic chemokine CCL5. In contrast, OPN may play an inhibitory role in adipogenic differentiation. Thus, the interplay between DCs and MSCs may contribute to the upregulation of OPN production with the consequent inhibition of MSC-derived adipogenesis and the induction of osteogenic differentiation [104,110].

Although most evidence on the crosstalk between MSCs and DCs has been retrieved from nondental tissue, a previous study highlighted that periapical lesion-derived MSCs (PL-MSCs) could differentially modulate DC functions and Th-cell response in periodontal lesions. At the beginning of the disease, PL-MSCs could activate resident DCs, inducing their maturation and Th1 polarization. In contrast, in the later stages of PLs, PL-MSCs could potentiate DC tolerogenic differentiation and the polarization of immune response toward Th2 and Treg cells, which are important for the healing stage of periodontal lesions. Thus, this study showed that PL-MSCs could either augment the inflammatory response or contribute to the resolution of the disease, depending on the dynamics of the two cell types [120].

The relationship between DCs and DPSCs/progenitor cells during the healing process was evaluated in vivo in two different studies using cavity preparation and tooth replantation mouse models. The most recent study in this topic elucidated the responses of the oral microflora-exposed dental pulp to capping with a triple antibiotic paste (TAP: metronidazole, ciprofloxacin, and minocycline) in mouse molars, compared to those to calcium hydroxide (CH) cement, in addition to the combination of macrogol and propylene glycol (MP, control group), followed by a glass ionomer cement filling. Quantitative real-time polymerase chain reaction (qRT-PCR) analysis was performed using specific primers for cDNA encoding *Nestin* (odontoblast differentiation), *Cd11c* (DCs), and *Nanog* (stem/progenitor cells). *Cd11c* mRNA sharply increased from weeks 1 to 2 in the TAP group, followed by the CH group, and correlated with a peak in the cell proliferative activity. In contrast, the MP group markedly decreased its expression 2 weeks after treatment, with significant differences between the TAP and MP groups regarding the mRNA expression of *Cd11c* on week 1. mRNA expression of the odontoblast differentiation marker *Nestin* was higher in the TAP group than in the CH and MP groups at week 1, followed by a steady decrease of its expression in all groups on week 2. TAP group also showed the highest mRNA expression of the stem cell marker *Nanog* among the three groups on week 1, whereas its expression decreased by week 2. These results demonstrated that TAP might contribute to a sterile environment that allows active pulpal cell proliferation and the simultaneous activation of DCs. Moreover, high mRNA expression of the stem cell marker *Nanog* on week 1 after treatment suggested that the use of TAP favored the activation of stem cells/progenitors residing in the injured dental pulp (Figure 7a) [37]. Likewise, another study evaluated the effectiveness of the combination of antibacterial drugs to heal the dental pulp. The maxillary first molars of 3-week-old mice were extracted and immersed in the 3Mix solution for 30 within compared to phosphate buffered saline (PBS) alone. Although no significant difference was found between the 3Mix and PBS groups, the study showed chronological changes in the gene expression of differentiation markers such as *Nestin*, *Dspp*, *Alp*, *Ocn*, and *Opn*, and the evaluation of DC (*Cd11c*) and DPSC (*Oct3/4A* and *B*) activity during the pulpal healing process. High expression levels of *Cd11c* mRNA were first observed in the 3Mix group on day 1 and later on days 5 and 7, whereas, in the PBS group, these levels progressively increased from days 1 to 7. Moreover, the *Oct3/4A* and *B* transcript leves were greatly enhanced on day 1 in both groups, correlating with the increase of *Cd11c* mRNA in the 3Mix group (Figure 7b).

Hence, immersion of replanted teeth in the 3Mix solution may have promoted DC migration in the dental pulp shortly after repositioning the tooth into the alveolar socket. The similar timing in the expression of DC and DPSC markers may involve hidden crosstalk between these two cell populations that might trigger stem/progenitor cell-mediated cell differentiation into odontoblast-like cells to begin the reparative dentin matrix deposition [36]. Further research is necessary to identify the possible crosstalk between DPSCs and pulpal DCs in the context of the odontoblast differentiation process after tooth injuries.

## 8. Conclusions

This literature review summarized the important findings related to the possible role of dental pulp DCs during physiological and pathological dentinogenesis retrieved from several in vivo and in vitro experiments performed over the last 20 years (Figure 8). Although the presence of DCs has been confirmed based on their morphological features after histological and immunohistochemical analyses, it is necessary to address the crosstalk between DCs and other important pulpal cells, such as odontoblasts and DPSCs, using further state-of-the-art in vitro strategies. To date, extensive research has been published elsewhere about the role of DCs toward inflammation caused by dental caries or trauma in the dental pulp; however, very little information is available on the molecular mechanisms that control the temporary appearance of DCs during the pulpal healing process. The lack of knowledge in this matter is particularly critical for establishing new strategies aimed to achieve the regeneration of the afflicted dental pulp tissue.

Since its first description 20 years ago, the concept of “osteoimmunology” has emerged as a field of study that unifies the complex interactions between bone biology and the immune system [121,122,123,124]. For instance, the study of bone pathologies, such as rheumatoid arthritis or osteoporosis, has shown us the great influence of the immune system on the function of bone-forming cells [122]. Both, skeletal and immune systems share several regulatory molecules, including cytokines, chemokines, receptors, and transcription factors. Therefore, this reciprocal relationship allows the achievement of critical bone functions, such as body support, regulation of mineral metabolism, and the hematopoietic process [124,125]. In addition, the study of this emerging discipline (osteoimmunology) is important not only to clarify the concepts related to its basic biology but also to develop new therapeutic strategies for bone and/or joint pathologies and immune disorders [123]. Hereby, the term “dentin-pulp immunology” is proposed in a similar fashion to “osteoimmunology.” The dentin and the pulp tissue share the same embryological origin. Both structures are derived from the ectomesenchyme of the maxillary and mandibular process of the first branchial arch and are considered as a functional unit [1]. Moreover, it is extensively proven that the dental pulp owns an innate immune system that allows the tissue response against oral pathogens or exogenous noxious stimuli. The complex network of immune cells that interplay with the pulpal cells triggers a phased natural repair process that varies according to the nature of the injury. The study of the biology of the dental pulp has shed light on the multiple functions of its inherent cells and the molecules critical for the execution of the main functions of the dentin-pulp complex. However, it is necessary to take a comprehensive approach to unify these concepts with the wide complexity of the immune system behind the healing responses.

Recently, immune-based therapies for the healing of the pulpal tissue, including the development of new immunomodulatory compounds for dental pulp regeneration or DPSC activation, are emerging as possible therapeutic alternatives in dentistry [126]. In this context, synthetic CpG-oligodeoxynucleotides (CpG-ODNs), have been shown to elicit strong host immunostimulatory responses through the sustained activation of APCs and are considered suitable adjuvants for treating cancer and infectious diseases [127,128]. Preliminary data from the authors suggested that the exposure to a CpG-ODN solution significantly improved the pulpal regeneration of replanted teeth in mice, possibly through the activation of pulpal DCs. Hence, a new treatment called “DC activation therapy” is also proposed, aiming to improve the prognosis of severely injured teeth toward the resolution of inflammatory conditions and their subsequent regeneration, as in avulsed teeth in children, or deep carious lesions with or without pulpal inflammation. Nevertheless, these promising in vitro results must be carefully analyzed as they do not contemplate the crosstalk with other cell populations and the influence of external stimuli. Further strategies should include using in vivo models and molecular and gene analyses of samples under experimental conditions to validate the potential therapeutic effects of new immunomodulatory compounds against pulp pathologies.

## Figures and Tables

**Figure 1 jcm-10-03348-f001:**
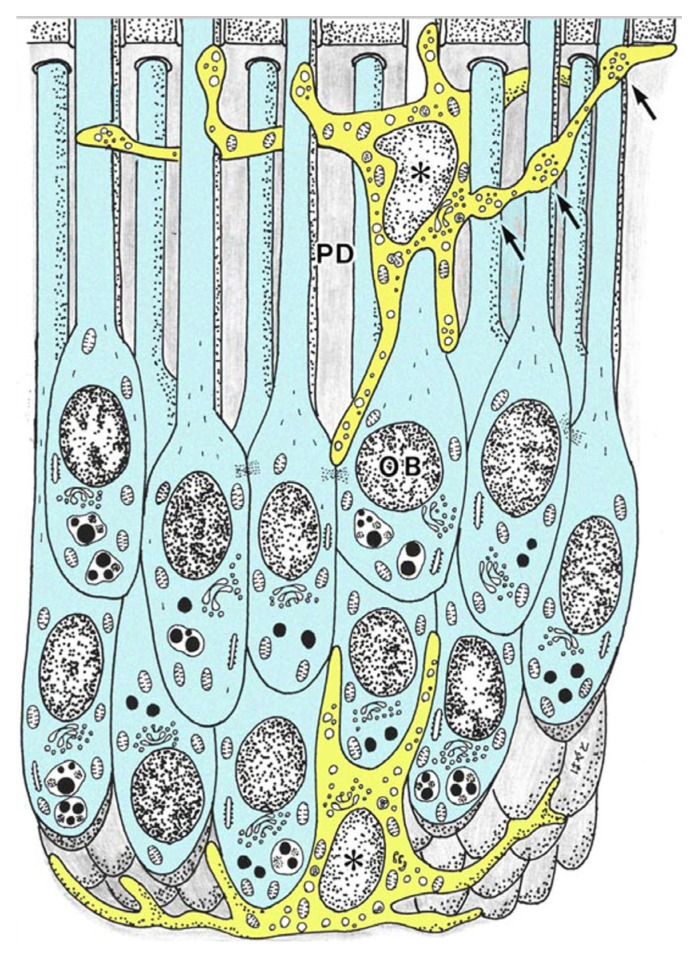
Schematic view of the spatial relationships between MHC^+^ class II cells and odontoblasts (OB) in the human dental pulp (cited from Ohshima et al. [82], with permission from Springer Nature). An MHC^+^ class II cell (asterisk) in the predentin (PD) makes contact with several odontoblast processes via cytoplasmic extensions that have a beaded appearance (arrows).

**Figure 2 jcm-10-03348-f002:**
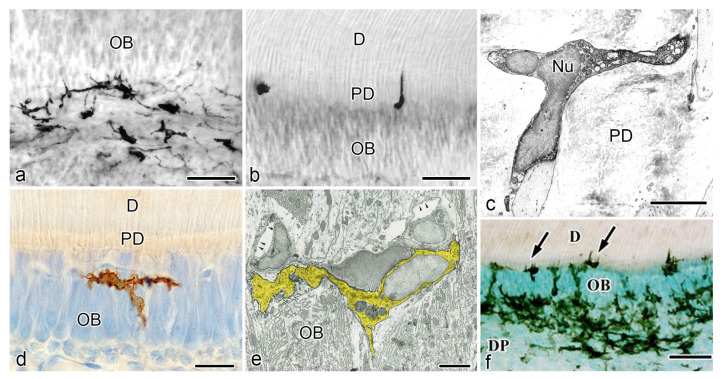
Relationship between odontoblasts and MHC^+^ class II cells during physiological dentinogenesis in human (**a**–**c**) and rat (**d**–**f**) pulp samples (cited from Ohshima et al. [82] with permission from Springer Nature and Ohshima et al. [39] with permission from Archives of Histology and Cytology). (**a**–**c**) Immunostaining for human leukocyte antigen (HLA)-DR antibody. (**d**–**f**) Immunohistochemistry with OX6 antibody. (**c**,**e**) TEM views. (**a**) Pulpal HLA-DR^+^ cells are identified at the periphery of the tissue, inside and beneath the odontoblast layer. (**b**) An HLA-DR^+^ cells inserting their cytoplasmic extension into the dentinal tubule. (**c**) Ultrastructure of an MHC^+^ class II cell in the predentin of human deciduous teeth. Clear and electron-dense vesicles of different sizes are seen in the cytoplasm. The number of organelles is scarce. (**d**) An MHC^+^ class II cell in the rat pulp exhibiting dendritic features. (**e**) Ultrastructural features of rat DCs. Multivesicular bodies, tubulovesicular structures, and a well-developed Golgi apparatus are seen in the cytoplasm of OX6^+^ cells. (**f**) OX6^+^ cells showing dendritic features in the dental pulp of aged rat molars. An intense immunoreaction is observed in the cellular bodies and cytoplasmic processes of OX6^+^ cells in the odontoblast and subodontoblastic layer. *D* dentin, *DP* dental pulp, *Nu* nucleus, *OB* odontoblasts, *PD* predentin, *Bar* = 50 μm (**a**,**b**,**f**), 20 μm (**d**), 4 μm (**c**), 3 μm (**e**).

**Figure 3 jcm-10-03348-f003:**
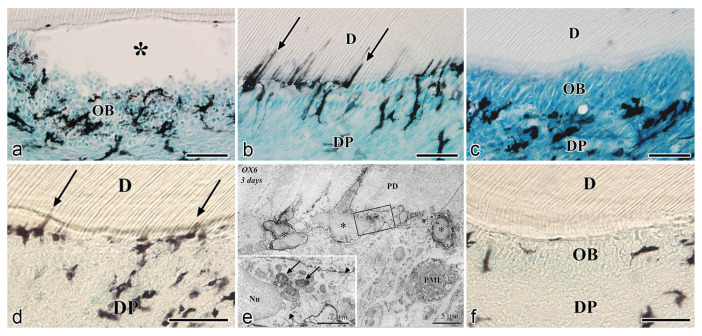
Relationship between DCs and odontoblasts or newly differentiated odontoblast-like cells after noninfected exogenous injuries in rat molars (cited from Ohshima et al. [44], with permission fromWilley and Nakakura-Ohshima et al. [45], with permission from Oxford University Press). (**a**–**c**) Cavity preparation model. (**d**–**f**) Tooth replantation model. (**a**–**f**) Immunohistochemistry with OX6 antibody. (**e**) TEM view. (**a**) Exudative lesions appear right after the initial injury beneath the injury site. OX6^+^ cells move inward from the pulp-dentin border and are relocated beneath the exudative lesions. (**b**) Twelve hours after cavity preparation, the number of OX6^+^ cells increase under the afflicted area (arrows). The exudative lesions disappear from 12 h to 1 day after the injury. (**c**) On day 3, the afflicted area of the pulp completely recovers. OX6^+^ cells are located exclusively beneath the odontoblast layer. (**d**) Three days after the tooth replantation, DCs located at the pulp-dentin border consistently express OX6 (arrows). (**e**) The ultrastructure of OX6^+^ cells on day 3 presents tubulovesicular structures and multivesicular bodies in their cytoplasm. An intense immunoreaction is observed in their cell membranes (arrows in inset). (**f**) OX6^+^ cells are observed in between or beneath the newly-differentiated odontoblast-like cells. *D* dentin, *DP* dental pulp, *OB* odontoblasts, *PD* predentin, *Bar* = 50 μm (**a**–**d**,**f**), 5 μm (**e**), 2 μm (inset). * exudative lesions.

**Figure 4 jcm-10-03348-f004:**
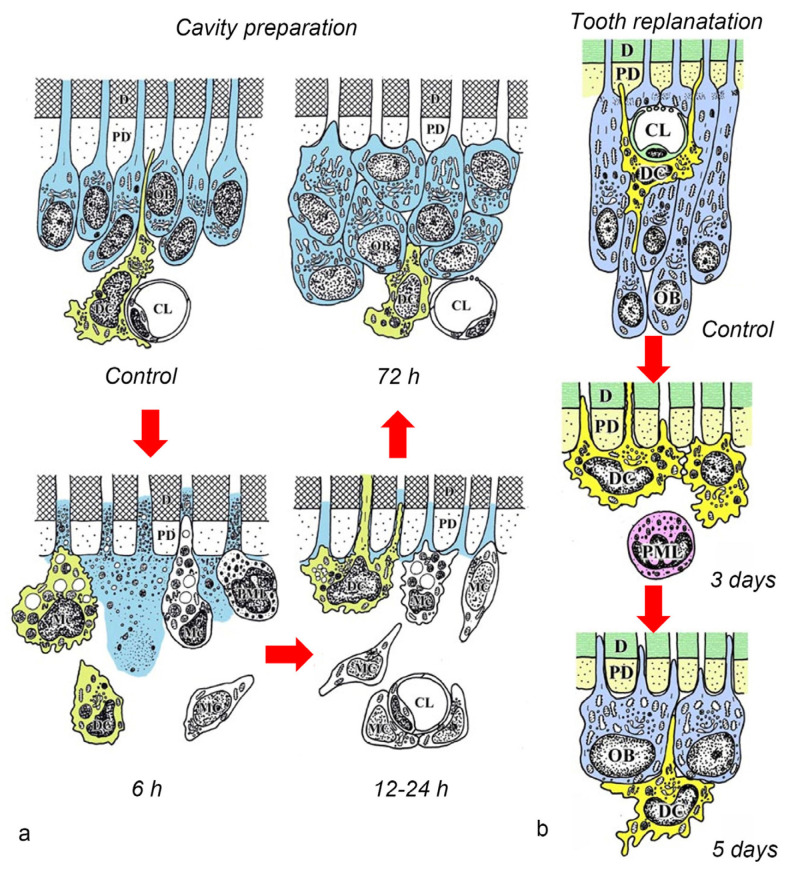
Schematic views of the morphological changes and spatiotemporal relationships between odontoblasts and DCs (OX6^+^ cells) during the early stages of the pulpal response in the cavity preparation (**a**) and tooth replantation (**b**) rat injury models (cited from Ohshima et al. [44], with permission from Wiley and Nakakura-Ohshima et al. [45], with permission from Oxford University Press). *CL* capillary lumen, *D* dentin, *Mc* macrophage, *MC* mesenchymal cell, *PD* predentin, *PML* neutrophile.

**Figure 5 jcm-10-03348-f005:**
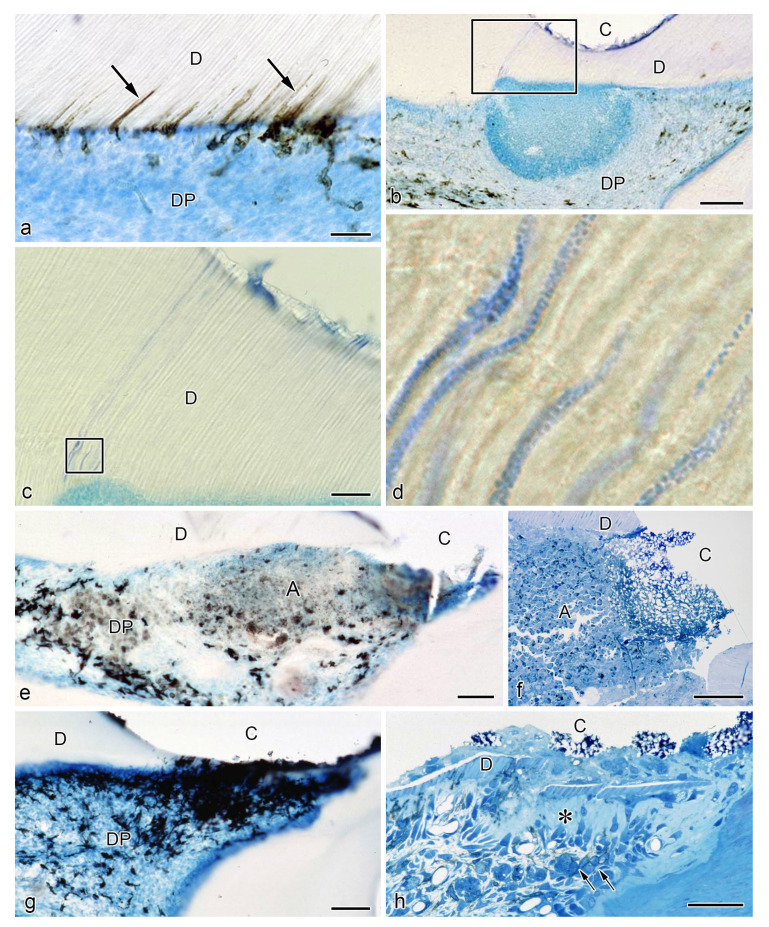
Responses of DCs to bacterial infection in the dental pulp. (**a**–**d**) Cavity preparation injury model using laser ablation (cited from Suzuki et al. [46], with permission from Springer Nature and Sato et al. [48], with permission from Archives of Histology and Cytology). (**e**–**h**) Cavity preparation and pulp capping model with αTCP cementum or αTCP cementum containing a 3Mix drug combination. (**a**–**h**) Immunohistochemistry with OX6 antibody. (**f**,**h**) Semithin sections. (**a**) OX6^+^ cells extend their processes deep into the dentinal tubules (arrows). (**b**) Laser ablation induces the infiltration of PMLs to form an abscess lesion in the dental pulp in most samples of the nontreated group from days 3 to 5 after the operation. Scattered OX6^+^ cells are observed at a distance from the abscess formation. (**c**,**d**) Higher magnification of the boxes in (**b**) and (**c**). The penetration of masses of oral bacteria can be identified in the dentinal tubules beneath the untreated cavity. (**e**) From day 3 to week 2 large abscess lesions, including numerous neutrophils, are still observed in injured teeth covered with αTCP cementum, particularly in the mesial half of the coronal pulp. (**f**) Neutrophil accumulation is observed at the exposure area covered with αTCP cementum. OX6^+^ cells are not consistently seen around the abscess lesions. (**g**) Numerous OX6^+^ cells accumulate along the pulp-dentin border of cavities treated with αTCP cementum containing a 3Mix drug combination. (**h**) Tertiary dentin deposition is accelerated in cavities covered with αTCP cementum containing a 3Mix drug combination. The newly formed dentin matrix is observed at the injury site (asterisk). OX6^+^ cells retreated beneath the odontoblast-like cells at this stage (arrows). *A* abscess, *C* cavity, *D* dentin, *DP* dental pulp, *Bar* = 100 μm (**b**,**e**,**g**), 50 μm (**f**,**h**), 25 μm (**a**,**c**).

**Figure 6 jcm-10-03348-f006:**
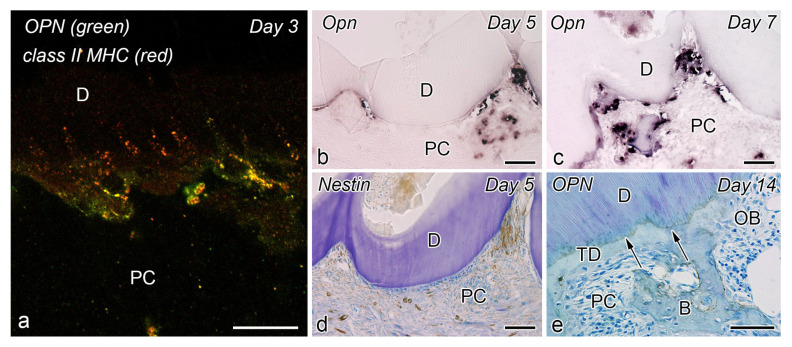
Relationship between OPN and DCs during pathological dentinogenesis (cited from Saito et al. [49], with permission from SAGE). (**a**) OPN and MHC class II double immunofluorescence staining. (**b**,**c**) OPN in situ hybridization and immunohistochemistry for (**d**) nestin and (**e**) OPN on day 3 (**a**), day 5 (**b**,**d**), day 7 (**c**), and day 14 (**e**) after tooth allogenic tooth transplantation. (**a**) MHC^+^ Class II cells (red) with dendritic features are also positive for OPN (green) on day 3 after transplantation, indicating that OPN^+^ cells are immunocompetent cells, such as macrophages and DCs. (**b**,**c**) Intense OPN reactions are seen at the mineralization front of the preexisting dentin on day 5. *Opn* mRNA^+^ cells are recognized at the pulp-dentin border in days 5 and 7. (**d**) Nestin-positive newly-differentiated odontoblast-like cells are arranged along the pulp-dentin border where OPN^+^ cells had previously existed. (**e**) Two weeks after the operation, OPN immunoreactions are observed at the boundary between the predentin and the newly-formed dentin matrix (arrows). OPN^+^ osteoblasts surround the bone matrix. *B* bone, *D* dentin, *OB* odontoblasts, *PC* pulp cavity, *TD* tertiary dentin, *Bar* = 100 μm (**c**,**d**,**e**), 50 μm (**e**), 10 μm (**a**).

**Figure 7 jcm-10-03348-f007:**
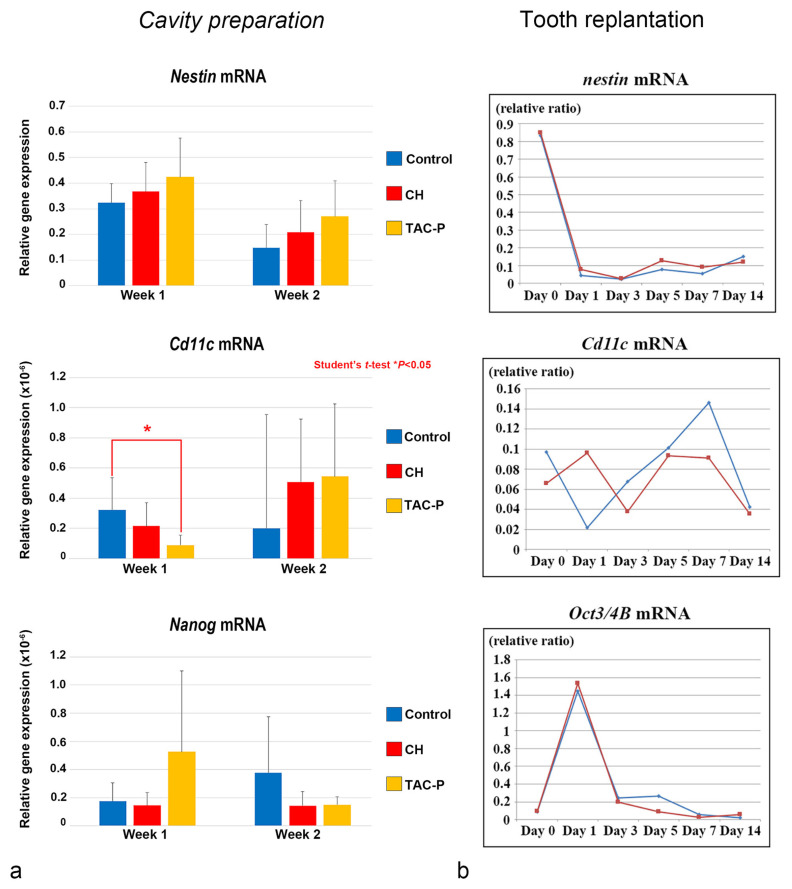
Relationship between DCs and stem/progenitor cells during the dental pulp healing process (cited from Quispe-Salcedo et al. [36,37], with permission from Elsevier). Dental pulp samples from (**a**) cavity preparation and (**b**) intentionally delayed tooth replantation models in mice. (**a**) qRT-PCR analysis for *Nestin*, *Cd11c*, and *Nanog* of dental pulp samples on weeks 1 and 2 after the cavity preparation and capping treatment with a triple antibiotic drug combination (3Mix). (**b**) qRT-PCR analysis for *Nestin*, *Cd11c* and *Oct3/4B* from day 0 to week 2 of dental pulp samples of intentionally delayed replanted teeth (30 min) treated with 3Mix solution. Student’s *t*-test * *p* < 0.05.

**Figure 8 jcm-10-03348-f008:**
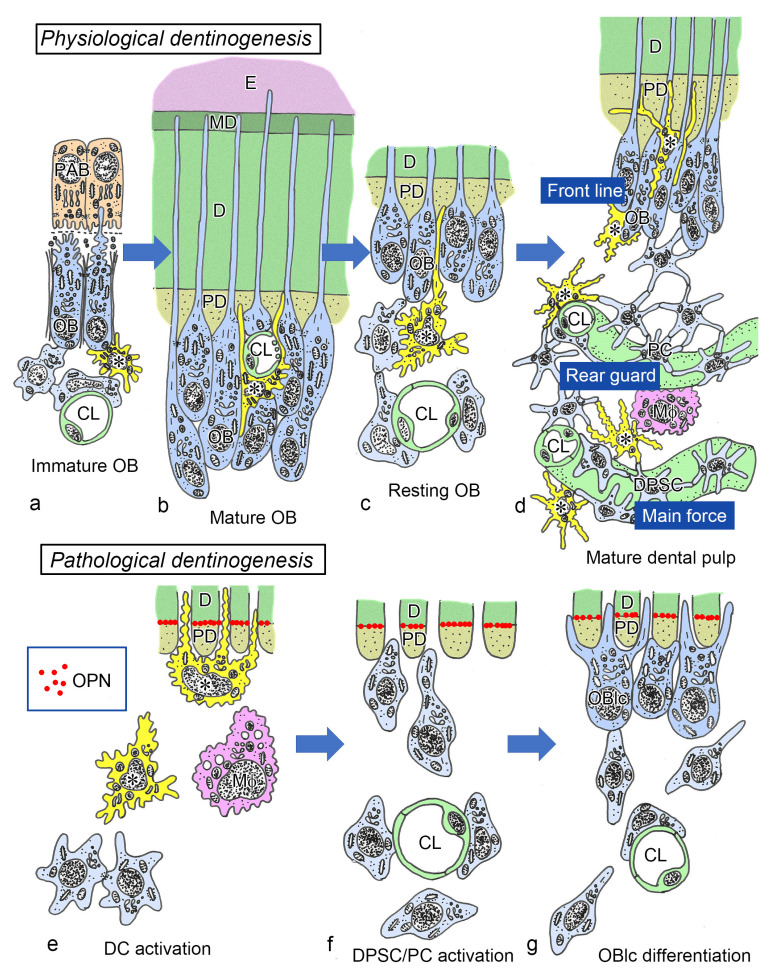
Schematic views summarizing the possible role of DCs (*) during physiological and pathological dentinogenesis. Osteopontin (OPN) is not, but the enamel epithelium and basement membrane are necessary for the odontoblast (OB) differentiation during physiological dentinogenesis. Odontoblasts are classified into immature (**a**), mature (**b**), and resting odontoblasts (**c**), and DCs associate with the odontoblasts in this process. Mature dental pulp (**d**) is composed of the frontline including odontoblasts, the rearguard including progenitor cells (PC), and the main force including dental pulp stem cells (DPSCs), and DCs associate with all compartments. OPN is deposited at the dentin-predentin interface during pathological dentinogenesis. OPN and DCs play a role in the sequential steps of DC activation (**e**), dental pulp stem/progenitor cell activation (**f**), and odontoblast-like cell (OBlc) differentiation (**g**). *CL* capillary lumen, *D* dentin, *E* enamel, *MD* mantle dentin, *M**φ* macrophage, *PAB* preameloblast, *PD* predentin.

## Data Availability

All data generated or analyzed during this study are included in this published article.

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
