# Peer review of "The Role of Dendritic Cells during Physiological and Pathological Dentinogenesis"

_jcm, 2021, doi:10.3390/jcm10153348_

Round 1

Reviewer 1 Report

This review paper is an excellent contribution and well written report on the function of dendritic cells in the dental pulp and during its healing processes to clinical challenges. It covers well the scientific progress over the years from the first description of this immunocompetent cell in the dental pulp more than 30 years ago to current issues on how the cell may interact with odontoblasts and dental pulp stem cells in healing and repair processes. At the end of the article emphasis is placed on the need to investigate the molecular mechanisms that guide the appearance of dendritic cells in healing processes. Authors further stress the need to include animal models to increase the knowledge on how new immunomodulatory compounds can be developed to enhance pulpal repair and regeneration processes. Over the years Hayato Ohshima has been a prominent author of research on dendritic cells in the dental pulp and in this reviewer’s opinion, publication of the article will be highly valuable to clinician-scientists having interest to advance the knowledge base in this area of pulp biology research.

Author Response

Thank you for reviewing our manuscript. We truly appreciate your positive and encouraging comments on our review. 

Reviewer 2 Report

Dear authors
It was an excellent opportunity to review this paper. It contained such broad and deep information about dendritic cells in pulp tissue. It is well organized, and I can notice a considerable effort by the authors.
Just a couple of minor suggestions are as follows.

1. Page 4, line 16: that sentence needs references.
2. Page 3, line 126: I think the title can be modified as follows "3. Relationship between DCs and odontoblast..." because the order of terms would be better to be consistent (DCs first and other cells or factors). This suggestion would be applied to another title  6. relationship between DCs and osteopontin.
3. Page 14, line 482-502: this paragraph needs references.

4. Lastly, I would like to suggest adding one or a couple of the authors' own digrams or tables that can summarize the paper.

Author Response

It was an excellent opportunity to review this paper. It contained such broad and deep information about dendritic cells in pulp tissue. It is well organized, and I can notice a considerable effort by the authors.

Just a couple of minor suggestions are as follows.

Thank you for your kind suggestions. The points of improvement are as follows.

  1. Page 4, line 16: that sentence needs references.

We totally agree with the reviewer’s notion. Although there is no line 16 in Page 4 (we believe there was an omission on the last digit of the line number), three suitable references were added in line 165, which probably was the sentence rising reviewer’s concern.

“Although studies using human teeth samples during the dentinogenesis process were pioneers and very informative, the data retrieved remained limited. Difficulties in sample acquisition and other ethical concerns led to the setting of murine models for further investigation in this subject and under pathological conditions [66-68].”

  1. Page 3, line 126: I think the title can be modified as follows "3. Relationship between DCs and odontoblast..." because the order of terms would be better to be consistent (DCs first and other cells or factors). This suggestion would be applied to another title 6. relationship between DCs and osteopontin.

We totally agree with the reviewer’s notion. The order of words in the titles was modified according to reviewer’s suggestions:

  1. Relationship between DCs and odontoblasts during physiological dentinogenesis
  2. 6. Relationship between DCs and osteopontin (OPN) during pathological dentinogenesis

  1. Page 14, line 482-502: this paragraph needs references.

 We totally agree with the reviewer’s notion. The missing references were added according to reviewer’s suggestion:

“The findings obtained using infection models demonstrate the importance of DCs as initiators of the pulpal immune response and suggest another putative role during the pulpal healing process [46,48,98,103]. The pulpal immune responses may vary according to the local conditions and the stimuli intensity, as the antigen-presenting capacity of pulpal DCs may persist for many months, even after caries treatment [83,100,102]. During the early stages after oral exposure, an active DC recruitment is described in the dental pulp under the exposure site [81,99]. Once an abscess lesion is formed, numerous neutrophils with phagocytotic activity occupy the afflicted odontoblast layer, whereas the remaining DCs migrate toward the center of the pulp and/or the surrounding areas. Changes in DC distribution may be related to the normal transit from the infection site to the regional lymphatic nodes for their maturation and the activation of the innate tissue-specific response of the dental pulp. This phenomenon does not occur if the infection is prevented by either covering the exposed cavity or using special filling materials, such as αTCP cement containing 3Mix drugs, for the in situ disinfection of the pulpal microenvironment. Although the signaling pathway connecting odontoblast-like cells and pulpal DCs is still unknown, it is evident that the interplay of odontoblasts and immunocompetent cells influences the healing of the pulpal tissue even under infection conditions, as all presented studies have shown that the number of OX6+ cells consistently decreased after tertiary dentin deposition [46,48,81,83,96-103]. Further research should confirm whether DCs play dual roles in both the initial immune response and the differentiation process of odontoblast-like cells under pathological conditions.”

  1. Lastly, I would like to suggest adding one or a couple of the authors' own diagrams or tables that can summarize the paper.

We agree with reviewer’s comment about the necessity of a diagram summarizing the findings of our review manuscript. We added an original diagram (Fig. 8).

Thank you again for your criticism.